# Optimal Validated Multi-Factorial Climate Change Risk Assessment for Adaptation Planning and Evaluation of Infectious Disease: A Case Study of Dengue Hemorrhagic Fever in Indonesia

**DOI:** 10.3390/tropicalmed7080172

**Published:** 2022-08-08

**Authors:** Lia Faridah, Djoko Santoso Abi Suroso, Muhammad Suhardjono Fitriyanto, Clarisa Dity Andari, Isnan Fauzi, Yonatan Kurniawan, Kozo Watanabe

**Affiliations:** 1Parasitology Laboratory, Biomedical Laboratory Group, Faculty of Medicine, Universitas Padjadjaran, Bandung 40161, Indonesia; 2Parasitology Division, Department of Biomedical Science, Faculty of Medicine, Universitas Padjadjaran, Bandung 45363, Indonesia; 3Infection Study Center, Faculty of Medicine, Universitas Padjadjaran, Bandung 45363, Indonesia; 4Graduate School of Science and Engineering, Ehime University, Matsuyama 790-8577, Japan; 5Climate Change Center, Institut Teknologi Bandung, Bandung 40132, Indonesia; 6Center for Marine Environmental Studies, Ehime University, Matsuyama 790-8577, Japan

**Keywords:** dengue risk assessment, least-square fitting, validation optimization, risk sensitivity, adaptive capacity

## Abstract

(1) Background: This paper will present an elaboration of the risk assessment methodology by Deutsche Gesellschaft für Internationale Zusammenarbeit GmbH (GIZ), Eurac Research and United Nations University Institute for Environment and Human Security (UNU-EHS) for the assessment of dengue. (2) Methods: We validate the risk assessment model by best-fitting it with the number of dengue cases per province using the least-square fitting method. Seven out of thirty-four provinces in Indonesia were chosen (North Sumatra, Jakarta Capital, West Java, Central Java, East Java, Bali and East Kalimantan). (3) Results: A risk assessment based on the number of dengue cases showed an increased risk in 2010, 2015 and 2016 in which the effects of El Nino and La Nina extreme climates occurred. North Sumatra, Bali, and West Java were more influenced by the vulnerability component, in line with their risk analysis that tends to be lower than the other provinces in 2010, 2015 and 2016 when El Nino and La Nina occurred. (4) Conclusion: Based on data from the last ten years, in Jakarta Capital, Central Java, East Java and East Kalimantan, dengue risks were mainly influenced by the climatic hazard component while North Sumatra, Bali and West Java were more influenced by the vulnerability component.

## 1. Introduction

In an area where Aedes and Culex mosquitoes proliferate, the burden of vector-borne diseases is very high. These diseases result in an immense loss in economies and restrict development in the urban and rural areas. Dengue is the most rapidly spreading mosquito-borne viral disease [1,2]. It is one of the neglected tropical diseases prioritized by the World Health Organization, with an estimated aggregated global cost of USD 8.9 billion in 2013 [2].

The transmission cycle of dengue viruses involves Aedes mosquitoes as the vector. The spread of dengue may develop into a public health emergency that will concern international health security. Reported case fatality rates for dengue are approximately 1%, but in the rural area of India, Indonesia and Myanmar, the case fatality rates are 3–5% [1,2]. The Indonesian Ministry of Health reported the highest case in 2016 with 201,885 cases, while the recent data in 2019 reported 138,127 cases. West Java contributed the most, with more than 20,000 cases—equal to 17% of the nation’s cases in 2019 [3,4].

Environmental and climate conditions, host–pathogen interactions, population immunological factors and urbanization influence dengue transmission [1,5,6,7,8]. Since there are no drugs or vaccinations to stop dengue, preventing or reducing dengue transmission relies on controlling the mosquito vector [1,9]. Mosquito vector control would continue to reduce the risk and burden of dengue, even after vaccine deployment [10]. Climate directly influenced the biology of the dengue vectors and their abundance and distribution [11].

Climate change impacts the intensity and risk of dengue in vulnerable communities [10,12]. It may change the burdens of dengue globally, nationally and locally [13]. It is necessary to develop spatial databases and link this information with related factors for a given area to better understand dengue distribution in terms of time and space [5]. Global risk maps developed for dengue could examine some of the scientific practices used to make predictions and decision-making [14].

The climate forecasting model of dengue for early warning systems based on climate variabilities and changes generally uses monthly, weekly, or daily climate data to obtain precise high prediction accuracy [15,16,17,18,19,20]. However, these specific data are quite laborious to collect nationwide, especially in geographically large countries such as Indonesia.

Moreover, although climate variables strongly influence dengue and its vectors, other elements affect disease occurrence [21]. The Indonesian government has built dengue programs and incorporates vector control, public health campaigns and education, training and research and epidemiological surveillance to counter the burden of dengue infection [22]. In a large country such as Indonesia, dynamic, accessible and easy-to-use technology may be a tool to deliver clinical and epidemiological data from the health sector and other relevant sectors to obtain information for health-related decision-making.

The Climate Change Risk and Adaptation Assessment (CCRAA) proposed by Deutsche Gesellschaft für Internationale Zusammenarbeit GmbH (GIZ), Eurac Research and United Nations University–Institute for Environment and Human Security (UNU-EHS) (2018) is a multi-dimensional evaluation system for adaptation planning at the national level [23]. This model is a standardized approach to climate risk assessments in the context of Ecosystem-based Adaptation (EbA) that provides relevant information on the climate-related risks of societies, economies and ecosystems, along the dimensions of hazard, exposure and vulnerability [23].

In this paper, we validate the dengue prediction model used for CCRAA by best-fitting it with the number of dengue cases in Indonesia using least-square fitting with a yearly data collection from the past ten years (2010–2019). This paper aims to elaborate on the CCRAA for dengue infectious diseases with a multi-dimensional evaluation system for adaptation planning in the health sector at the national level. The variations of potential climate change risk on the dengue infectious diseases were validated and adjusted by an optimal technique of reconciliation with evidence of incidence numbers (provincial).

## 2. Materials and Methods

### 2.1. Climate Change Risk and Adaptation Assessment

The methodology of CCRAA refers to the guidebook on Climate Risk Assessment developed by GIZ, which was then detailed and implemented by the Overseas Environmental Cooperation Center (OECC), Japan in the title of vulnerability and risk pluralistic evaluation system (VULPES) [23,24]. This, in principle, represents a spatial risk distribution of climate change potential impacts based on the Fifth Assessment Report of the Intergovernmental Panel on Climate Change (IPCC AR5) [25].

In Indonesia, CCRAA was outlined per the Regulation of the Ministry of Environment and Forestry (KLHK) No 33 of 2016, which is based on the Guideline of CCRAA developed by the Ministry of Environment in 2012 [26]. In early 2021, a research collaboration between the Climate Change Centre Institut Teknologi Bandung (CCC-ITB) and the Japanese Ministry of Environment (MoEJ) refined the CCRAA methodology through the elaboration of climate change impact chain in different sectors of national development, including the health sector. The CCRAA of dengue was conducted by development of impact chain and risk assessment [23]. This paper will focus on dengue cases as an indicator of the health sector.

#### 2.1.1. Development of Impact Chain

The impact chain (cause–effect chain) is an analytical tool to help better understand, systemize and prioritize the factors that drive risk in the system of concern (dengue). Impact chain was developed prior to risk assessment by determining the three risk components of hazard, exposure and vulnerability components [23,26]. The climatic hazard is defined as potential climate-related physical events that may cause loss of life, injury, or other health impacts. Therefore, climatic hazard selected components were the data of temperature, precipitation, relative humidity and El Niño–Southern Oscillation (ENSO). Exposure is defined as the presence of people and livelihoods in places and settings that could be adversely affected by the impacts, hence the population density data were selected as the exposure components. Vulnerability, which includes sensitivity and adaptive capacity components, represents the sensitivity to harm and to what extent the lack of capacity to cope and adapt. Sensitivity data selected were the number of the vulnerable population, the ratio of poor people and the number of villages near the river, while adaptive capacity data selected were collected from the available data from the yearly publication of the Indonesian Health Profile by the Indonesian Ministry of Health. The result of the development of the impact chain diagram was drawn up in Figure 1.

#### 2.1.2. Risk Assessment

As we involved different types of risk component data with various dimensions and units, the acquired data were normalized with a scale from 0 (optimal) to 1 (critical) by the normalization equation [23,27]:(1)$$X^{\prime }=\frac{X-X_{min}}{X_{max}-X_{min}}$$

Note: *X’* = normalized value; *X* = a set of the observed values present in *X*; *X_min_* = the minimum values in *X*; *X_max_* = the maximum values in *X*.

Normalization converts numbers into meaning by evaluating the criticality of an indicator value with respect to the risk [23]. For components with factors that are inversely proportional to increased risk, the normalization formula was adjusted to become [24]:(2)$$X^{\prime }=1-\frac{X-X_{min}}{X_{max}-X_{min}}$$

The min–max normalization formula uses the minimum and maximum data from all the seven provinces (all min–max data are the same for each province). To combine the normalized indicators into a composite indicator representing a single risk component (hazard, vulnerability and exposure), each data risk component was aggregated with weighted linear combinations according to the conditions in the study area Equation (3) [23].
(3)$$CI=\frac{\left(I_{1}\;x\;w_{1}\right)+\left(I_{2}\;x\;w_{2}\right)+\ldots +\left(I_{n}\;x\;w_{n}\right)}{w_{H}+w_{V}+w_{E}}$$

Note: *CI* = composite risk component indicator, e.g., hazard; *I*_n_ = individual indicator of the risk components, e.g., precipitation; $w_{n}$ = weight assigned to the individual indicator. 

Risk assessment in principle follows a step in GIZ, EURAC and UNU-EHS (2018) that involves formulation of physical risk as a function of hazard and vulnerability components in a particular exposure area [23]. These three risk components of hazard, vulnerability and exposure become one single composite risk indicator. We choose a one-step approach using the weighted arithmetic mean to simplify our model validation, which remains consistent with the Intergovernmental Panel on Climate Change Fifth Assessment Report risk concept (IPCC AR5) [25]. The risk is aggregated from its component by weighted arithmetic aggregation (linear combination) as follows [23]: (4)$$Risk=\frac{\left(Hazard\;x\;w_{H}\right)+\left(Vulnerability\;x\;w_{v}\right)+\left(Exposure\;x\;w_{E}\right)}{w_{H}+w_{V}+w_{E}}$$

Note: $w_{H}$ = weighted arithmetic mean of hazard; $w_{v}$ = weighted arithmetic mean of vulnerability; $w_{E}$ = weighted arithmetic mean of exposure.

### 2.2. Risk Components/Indicators and Data Requirements

For the model development, we used the data based on the impact chain, which are the number of dengue cases, climate data for hazard components, as well as statistical data on population and health sector for vulnerability components, as described below. All of the data were collected over the last ten years, from 2010 until 2019. Incomplete data were filled in by using a proxy and reviewed based on the considerations from the expert (see Appendix A, Table A1). All components of data are summarized in Table 1.

#### 2.2.1. Climatic Data Collection for Hazard Components

Climatic data were collected from some public accessed databases as follows:The temperature of the earth’s surface was collected from the fifth generation European Centre for Medium-Range Weather Forecasts (ECMWF) atmospheric reanalysis of the global climate, ERA5 [28];Precipitation was collected from the Jaxa Global Rainfall Watch [29];Relative humidity was calculated from surface and dewpoint temperatures by the equation from Lawrence (2005) [30], where the respective data were collected from ERA5 [28];
(5)$$RH=100-\left(5\times \left(t-t_{d}\right)\right)$$

Note: *RH* = relative humidity; *t* = temperature of the earth surface; $t_{d}$ = 2 m dewpoint temperature.

4.The Oceanic Nino Index was collected from the National Weather Service Climate Prediction Center [31].

#### 2.2.2. Population Data Collection for Exposure and Vulnerability (Sensitivity) Components

Population density, the number of vulnerable populations (productive age population: 15–64 years old), the number of villages near rivers (the counted villages are located on both sides of the riverbed which is calculated from the edge to the foot of the inner embankment) and the ratio of poor people (the number of people who have an average monthly per capita expenditure below the poverty line at that time), were obtained from the Indonesian Central Bureau of Statistics’ official website [32]. Indonesian Central Bureau of Statistics is a non-ministerial government institution directly responsible to the President of Indonesia.

#### 2.2.3. Health Sector Data Collection for Vulnerability (Adaptive Capacity) Components

Health sector data were collected from the Indonesian Ministry of Health’s official website [3,4,33,34,35,36,37,38,39,40]. This is a yearly publication of “Profil Kesehatan Indonesia”, an Indonesian Health Profile. Data collected were the density and distribution of health workers, the number of public health centers, the number of hospitals, the number of general practitioners, the number of villages with health centers, the spending of the health deconcentration fund (tied grants from the Ministry of Health to be used for centrally specified sectoral activities), percentage of health deconcentration fund spending (and the number of health facilities of BPJS (Health and Social Security Agency).

### 2.3. Risk Level/Categorical Class Value as a Guide to Adaptation Actions

The decision for adaptation actions (including planning and evaluation) usually requires prioritization, which includes areas or stakeholders to receive more resources and otherwise, depending on the magnitude of risks. Therefore, the governments could consider the adaptation actions based on the risk level or risk assessment categorical class (see Table 2).

### 2.4. Scoping Study Area by the Case Incidence Data 

Among 34 provinces of Indonesia, this study focused on 7 out of the 34 of Indonesia’s provinces [41]. The provinces selected were North Sumatra, Jakarta Capital, West Java, Central Java, East Java, Bali and East Kalimantan (see Figure 2) [3,4,33,34,35,36,37,38,39,40]. The selection of the 7 provinces is based on the classification of the average dengue cases incidence (see the data in Appendix B, Table A2) into 5 levels by the natural break methods in QGIS application. The seven provinces are the ones with Very High (red) and High (orange) levels of dengue cases number. They also have high standard deviations of dengue cases, indicating an influence of climate variabilities over a yearly period.

### 2.5. Optimal Validation of Climate Change Risk Assessment Model

Validation of a model is a set of processes and activities intended to determine to what level of accuracy the model is developed representing the underlying real system being modeled. According to Keijnen (1999), validation is an effort on determining whether the simulation model is an acceptable representation of the real system—given the purpose of the simulation model [42].

We validate the model of climate change risk assessment of the infected disease by the approach of best-fitting the model with the numbers of dengue incidence as evidence-based impact. Some approaches could be applied to estimate weights of respective risk components, e.g., similarity, nonlinear optimization, or even qualitative expert judgment, in terms of validation of the method toward the data representing current or past impacts. For this paper, we used the least-squares method to estimate optimal weights (applied to minimize the difference between the normalized risk value and the number of dengue cases that were normalized as well). This method was chosen since this method could minimize the difference between the results of the climate change risk assessment and the data on the number of dengue cases in Indonesia collected from the Indonesian Ministry of Health [3,4,33,34,35,36,37,38,39,40].

The least-squares method determines the best fit line for the data by using simple calculus and linear algebra as the proof. The linear combination is [43]:(6)$$y=\alpha_{1}f_{1}\left(x\right)+\ldots +\alpha_{k}f_{k}\left(x\right)$$

The least-square fitting was applied optimally by using a smooth nonlinear minimization algorithm of Generalized Reduced Gradient (GRG) Nonlinear as an add-in in Excel Solver. The method searches for gradient or slope of least-square fitting as the objective function and determines that it has achieved an optimum solution when the partial derivatives equal zero [44].

The minimization of the differences between the risk and the number of dengue cases was estimated by [43]: (7)$$E\;\left(a,b\right)=\overset{N}{\underset{n=1}{\sum }}{\left(y_{n}(\alpha x_{n}+b)\right)}^{2}$$

Based on the results of least-square fitting, we produced the weighted arithmetic mean of each component based on the evidence of the yearly number of dengue cases (Appendix C, Table A3).

## 3. Results

### 3.1. Optimal Weight Estimation by Validation of Risk Assessment Model

The results of the weight estimation were different among the provinces and for each risk component (Appendix C). The difference in estimation reflects each province’s different conditions, as the value was validated by the yearly number of dengue cases. The optimal weight estimation by validation of the risk assessment model minimized the difference between the results of the risk assessment and the data on the yearly number of dengue cases (Figure 3).

### 3.2. Profiles of the Risk and Its Components

A risk assessment was carried out in Equation (4) using the weights from Appendix B with the normalized aggregate data of hazard, exposure and vulnerability. The results of the risk assessment for each province can then be seen in Table 3. The risk profile as the assessment results based on data from the last ten years showed that six of the seven provinces analyzed have an average, medium-risk level (risk = 0.41–0.60) of dengue cases. One province showed low-risk results—Bali Province.

Based on the three risk factors (hazard, exposure, vulnerability), the vulnerability component can be optimized by policyholders. Meanwhile, the hazard factor can be used as an early warning system in terms of the development planning in the health sector, projections of which are not discussed in this paper. The system could be used as a science-based intervention tool for developing yearly health planning and evaluations that are influenced by climate change impacts. This system is different from the WHO early warning systems [45,46,47] since this system not only relates climate to the number of cases but also includes exposure and vulnerability variables (including sensitivity and adaptive capacity components) in the climate assessment. Which, based on the results of the study analysis, shows that it has a role in risk (Figure 4).

Figure 4 shows that the dengue risk in several provinces was mainly influenced by the climatic hazard component (red color), in East Kalimantan, Jakarta Capital, Central Java and East Java. Meanwhile, in North Sumatra, Bali and West Java, the dengue risk was more influenced by the vulnerability component (blue color).

Furthermore, a combination of hazard and vulnerability component results could be used as an evaluation system for the implementation of adaptation planning or health intervention. For provinces where the number of dengue cases was affected mainly by the vulnerability risk component (North Sumatra, Bali and West Java), it is possible to identify which vulnerability components can be adapted, especially the vulnerability-adaptive capacity component. Table 4 shows the normalized and weighted data of the vulnerability components. The data show that the vulnerability components that most influenced the risk of the number of dengue cases in the seven provinces in Indonesia are V4 (number of hospitals) and V3 (number of public health centers).

## 4. Discussion

The climatic hazard data chosen were temperature, precipitation, relative humidity and ENSO. This was based on the research by Hasanah and Susana (2019), which showed a significant relationship between the weather variables above and the dengue fever cases in the Jakarta Capital from 2008 to 2016 [48]. Furthermore, research by Arcari and Tapper (2017) in Indonesia showed that indicators of ENSO assist in the forecast of potential dengue incidence and distribution in Indonesia [49]. Therefore, we added ENSO as the fourth climatic hazard data in this study. For the exposure component, we used the population density data. This is based on spatial analysis in a regency (Kabupaten) in South Kalimantan province, Indonesia, that showed that the highest dengue cases occurred in an area with the highest population density in the province [50]. Furthermore, it is also confirmed that there is an association between population density, vector production and dengue transmission [6].

The vulnerability (sensitivity) data collected were the number of the vulnerable population (productive age), the ratio of poor people and the number of villages near the river. The number of vulnerable populations was collected based on the findings of Utama et al. (2019) and Fuadzy et al. (2020), who showed that dengue virus infection in Indonesia tended to affect productive age patients and affect the increase in dengue cases significantly [51,52]. The ratio of poor people was chosen since Nuryunarsih (2015) found that population density and poverty had a significant correlation to dengue cases [53]. The number of villages near the river was chosen based on the Kusumawati et al. (2016) study which predicted that increased river flows will affect the increase in dengue cases in Sukoharjo, Indonesia [54]. This is supported by Hsueh et al. (2012) who confirmed the importance of the role that water bodies played in the spread of dengue fever in Taiwan and Hashizume et al. (2012) who reported both high and low river levels increased the hospitalizations of dengue fever cases in Dhaka, Bangladesh [55,56].

Vulnerability (adaptive capacity) data were collected from the available data from the yearly publication of the Indonesian Health Profile by the Indonesian Ministry of Health [3,4,30,31,32,33,34,35,36,37]. The chosen data could be adapted by the Indonesian Government to decrease the dengue cases in Indonesia (Table 1). All data components used in this study can be changed according to expert considerations and the needs of each related party. This is very useful for Indonesia where the Health Information System faces several major obstacles [22]. This method makes it easier for the related parties with limited data access but requires a decision in conducting an analysis of adaptation planning and evaluation in their respective fields. Because the results will be followed with the entered variables only, the role of each field and variable in the desired risk outcome share can be estimated.

Furthermore, the optimal determination of the weight value in the risk assessment was carried out by validating the model by best-fitting it with the number of dengue cases using the least-square fitting method. The number of diagnosed dengue cases was selected as the validation of the risk since it is useful for measuring the impact of dengue on health services and for their further analysis such as prevalence, cumulative incidence and incidence rate [57]. This causes our research to be unique from other analyses since the optimal weight value is based on the evidence of the dengue number of cases in each province and each year. We are convinced this weighting is appropriate for Indonesia, which is an archipelagic country with a vast area and varied geography, high biodiversity, population densities and characteristics [58].

Analysis for the past ten years showed that North Sumatra, Jakarta Capital, West Java, Central Java, East Java and East Kalimantan tended to be in the same risk category class—medium-risk (risk = 0.41–0.60). Meanwhile, Bali was in the low-risk category class with 0.35. These results indicate that the risk of dengue fever in all seven provinces was likely, except for Bali, where the results were lower and fall into the low-risk category class. Therefore, the seven provinces with the highest average of dengue cases in 2010–2019 had similar risks with the exception of Bali, which had a lower risk. Data normalization causes the results of the risk calculation to be connected. If all Indonesian provinces are included, different risk levels will be obtained. Therefore, further assessments are possible to compare all 34 provinces or other selected administrative areas when setting priorities for adaptation planning and evaluation of infectious diseases.

Because the risk model was validated with data on the number of dengue cases using best-fitting, the risk data in Table 3 correspond to the state of the number of dengue cases in Indonesia. The number of dengue cases was higher in 2016, 2015 and 2010, which might have been caused by La Nina in 2010, the strong El Nino in 2015 and the weak La Nina in 2016, respectively, which influence the relative humidity, wind and rainfall in Indonesia [59,60,61,62,63].

Figure 4 shows that there were differences in the risk components among the provinces. East Kalimantan, Jakarta Capital, Central Java and East Java were mainly influenced by the climatic hazard component (red color). Meanwhile, North Sumatra, Bali and West Java were more influenced by the vulnerability component (blue color). This result was in line with the risk analysis of North Sumatra, West Java and Bali, which tends to be lower than the other provinces in Table 3 in 2010, 2015 and 2016 because the influence of climate was not high in these three provinces while other provinces have a risk that tends to be higher.

Provinces with the risk of dengue fever cases were mainly influenced by the vulnerability component, North Sumatra was influenced by V3 (number of public health centers per city or district), West Java was influenced by V3 and V4 (number of hospitals per city or district), while Bali was influenced by V4 and V10 (percentage of health deconcentration fund spending) (see Table 4). The difference in adaptive capacity per province shows that local governments can intervene according to the needs of their respective regions. Nationally, the Indonesian government can consider the adaptive capacity that has the most influence on risk as a priority in the National Action Plan for Climate Change Adaptation (Rencana Aksi Nasional Adaptasi Perubahan Iklim, RAN-API). The results of this study can be used by the Indonesian Ministry of National Development Planning (BAPPENAS) as an input for their short- and long-term plans. 

Based on the results of the analysis, each local government can carry out adaptation planning and evaluation of dengue infectious disease by focusing on improving public health centers, hospitals, or health deconcentration fund spending. Public health centers and hospitals are important since they are responsible for dengue diagnosis, reporting and care [64,65]. Therefore, Indonesia needs to strengthen the dengue case reporting system and improve the health system and infrastructure at public health centers and hospitals [64,66], especially when extreme climate change occurs, such as the El Nino and La Nina events in 2010, 2015 and 2016.

Especially for Bali, the local government needs to focus on the health deconcentration fund spending for dengue. Based on Caballero-Anthon’s (2015) statement, there is an issue of sustainability and revolving funding sources for dengue interventions in Indonesia and this is also evident at the local government level [64]. Thus, local health coverage varies depending on the budget capacities and constraints. Caballero-Anthon also added that areas with high tourism rates such as Bali can leverage associated economic growth to collaborate with the private sector and mobilize communities to improve and sustain interventions at the local level [64].

The risk assessment by GIZ utilized in this study has its limitations and has the disadvantage that a positive value for one component can mask the importance of values for other components, unfavorably masking important issues in the system [23]. Furthermore, the linear least-squares method, which can assume long ranges, is not extrapolative and may be sensitive to outliers [67]. A limitation of this paper is that the data used are only from a duration of 10 years, from seven provinces, and cannot describe Indonesia as a whole. In the future, more specific data such as mosquito surveillance results, blood smears and the number of surveys by month could be used by the authorities concerned to conduct more a detailed analyses (e.g. using data on dengue cases in relation to age group, gender, etc.).

## 5. Conclusions

We demonstrated optimal validated multi-factorial climate change risk assessment for adaptation planning and the evaluation of infectious disease with a case study of dengue hemorrhagic fever in Indonesia. The health intervention as the adaptation actions could use the optimal vulnerability-adaptive capacity component or the exposure and vulnerability-sensitivity components for the impacts of climatic hazards. Risk assessment based on evidence of the number of dengue cases showed an increased risk in 2010, 2015 and 2016 in which the effects of El Nino and La Nina extreme climates occurred. This model showed the climate change effects to the risk of dengue incidence along with the vulnerability components (sensitivity and adaptive capacity), which showed a different value per province. North Sumatra, Bali and West Java are more influenced by the vulnerability component, in line with their risk analysis, which tended to be lower than the other provinces in 2010, 2015 and 2016 when El Nino and La Nina occurred. The vulnerability components that most influenced North Sumatra were the number of public health centers per city or district. West Java was influenced by the number of public health centers per city or district and the number of hospitals per city or district. Bali was influenced by the number of hospitals per city or district and the percentage of health deconcentration fund spending. This data can be used as a basis for further research as well as for policymakers and public health practitioners in the management of dengue.

## Figures and Tables

**Figure 1 tropicalmed-07-00172-f001:**
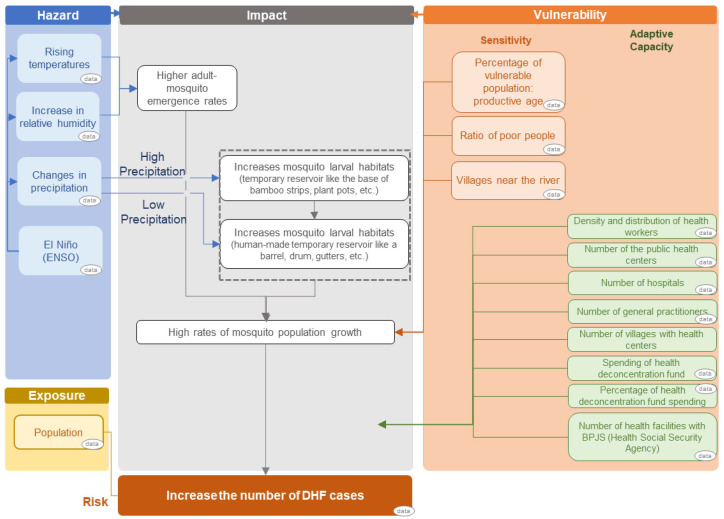
Results of climate change impact chain of the dengue number of cases.

**Figure 2 tropicalmed-07-00172-f002:**
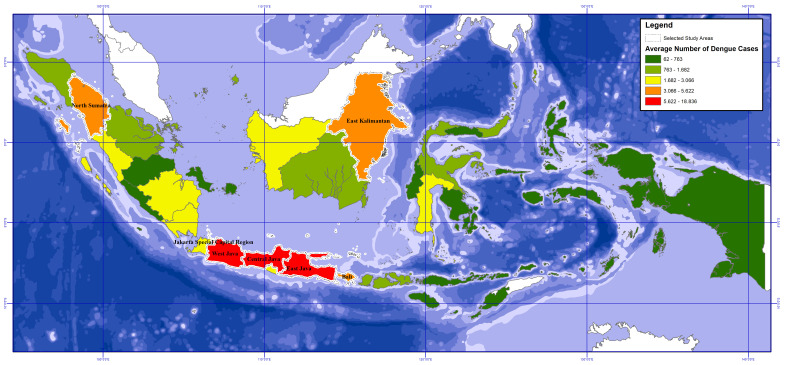
Map of the selected study areas based on the average number of dengue cases in various provinces.

**Figure 3 tropicalmed-07-00172-f003:**
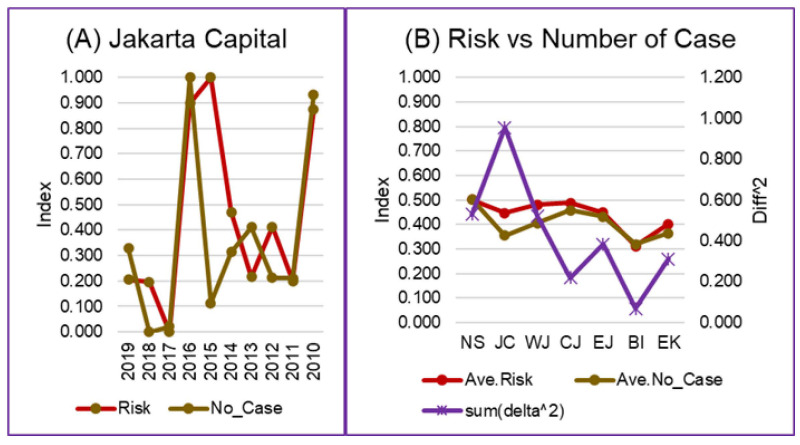
Risk and number of dengue cases comparison: (**A**) A comparison of the annual risk and number of dengue cases in the Province of the Jakarta Capital. (**B**) The comparison between the average risk and the average number of dengue cases in the seven analyzed provinces. Abbreviations: NS = North Sumatra, JC = Jakarta Capital, WJ = West Java, CJ = Central Java, EJ = East Java, BI = Bali, EK = East Kalimantan. (Graphs of the comparison of annual risk and number of dengue cases in other provinces can be seen in Appendix D, Figure A1).

**Figure 4 tropicalmed-07-00172-f004:**
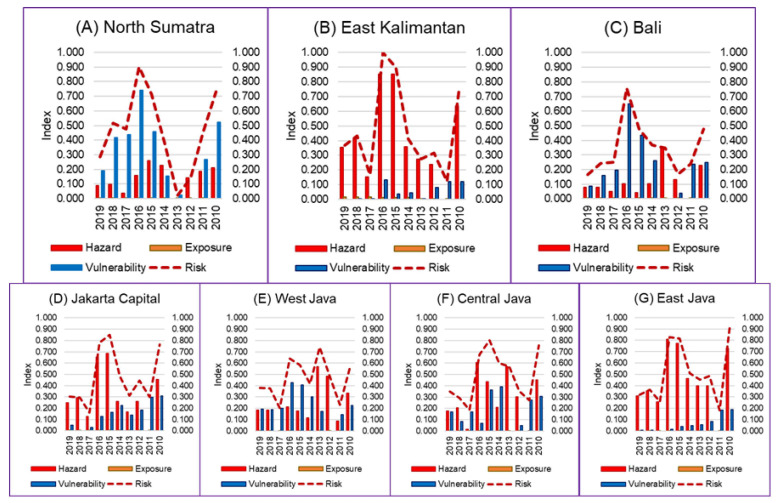
Dengue risk components of seven provinces in Indonesia: (**A**) North Sumatra Province, (**B**) East Kalimantan Province, (**C**) Bali Province, (**D**) Jakarta Capital Province, (**E**) West Java Province, (**F**) Central Java Province, (**G**) East Java Province.

**Table 1 tropicalmed-07-00172-t001:** Components of data collection.

Components (Year)	Collected Data	Unit
Hazard (2010–2019)	Temperature	°C
	Precipitation	mm/year
	Relative humidity	%
	El Niño–Southern Oscillation (ENSO)	scale
Exposure (2010–2019)	Population density	person/km^2^
Vulnerability-Sensitivity (2010–2019)	Number of vulnerable population (Age of 15–64)	person/population
	The ratio of poor people	%
	Number of villages near the river	unit/city or district
Vulnerability-Adaptive Capacity (2010–2019)	Density and distribution of health workers	person/city or district
	Number of the public health centers	unit/city or district
	Number of hospitals	unit/city or district
	Number of general practitioners	person/city or district
	Number of villages with health centers	village/city or district
	Spending on health deconcentration fund	rupiah/population
	Percentage of health deconcentration fund spending	%
	Number of health facilities with BPJS (Health Social Security Agency)	unit/city or district

**Table 2 tropicalmed-07-00172-t002:** Risk categorical class values by GIZ, EURAC and UNU-EHS (2018) [23].

Risk Level or Categorical Class Value (1–5)	Class Value (0–1)	Description
1	0.00–0.20	Very Low
2	0.21–0.40	Low
3	0.41–0.60	Medium
4	0.61–0.80	High
5	0.81–1.00	Very High

**Table 3 tropicalmed-07-00172-t003:** Dengue risk assessment for seven provinces in Indonesia.

Year	North Sumatra	Jakarta Capital	West Java	Central Java	East Java	Bali	East Kalimantan
2019	0.29	0.30	0.38	0.35	0.32	0.17	0.37
2018	0.52	0.30	0.38	0.29	0.37	0.24	0.43
2017	0.48	0.16	0.20	0.19	0.26	0.25	0.17
2016	0.90 **	0.78 *	0.64 *	0.68 *	0.83 **	0.76 *	1.00 **
2015	0.72 *	0.85 **	0.58	0.80 *	0.81 **	0.48	0.90 **
2014	0.38	0.49	0.42	0.60	0.51	0.37	0.41
2013	0.02	0.31	0.74 *	0.57	0.45	0.35	0.27
2012	0.14	0.45	0.48	0.35	0.48	0.17	0.32
2011	0.45	0.30	0.23	0.27	0.18	0.24	0.12
2010	0.73 *	0.77 *	0.56	0.76 *	0.92 **	0.48	0.76 *
Average	0.46	0.47	0.46	0.49	0.51	0.35	0.48
Description	Med	Med	Med	Med	Med	Low	Med

* High risk (0.61–0.80) ** very high risk (0.81–1.00).

**Table 4 tropicalmed-07-00172-t004:** Normalized and weighted data of the vulnerability components.

Province	V1	V2	V3	V4	V5	V6	V7	V8	V9	V10	V11
North Sumatra *	0.027	0.001	**0.764**	0.001	0.000	0.001	0.000	0.004	0.001	0.006	0.040
Jakarta Capital	0.000	0.000	0.005	0.099	0.000	0.001	0.000	0.000	0.024	0.033	0.016
West Java *	0.018	0.002	**0.279**	**0.153**	0.000	0.000	0.000	0.000	0.000	0.013	0.000
Central Java	0.000	0.001	0.097	**0.464**	0.001	0.000	0.000	0.002	0.085	0.000	0.058
East Java	0.002	0.001	0.087	**0.632**	0.001	0.000	0.000	0.002	0.009	0.000	0.013
Bali *	0.002	0.001	0.096	**0.162**	0.001	0.001	0.000	0.000	0.011	0.106	0.043
East Kalimantan	0.000	0.001	0.029	**0.870**	0.001	0.001	0.000	0.001	0.003	0.002	0.011

* Province more influenced by the vulnerability component. The bolded numbers are the components that are relatively higher than other vulnerability components. Abbreviations: V1 = proportion of the vulnerable population, V2 = density and distribution of health workers per city or district, V3 = number of public health centers per city or district, V4 = number of hospitals per city or district, V5 = number of general practitioners per city or district, V6 = number of villages with health center per city or district, V7 = number of villages near the river per city or district, V8 = percentage of poor people, V9 = proportion of the spending of health deconcentration fund, V10 = percentage of health deconcentration fund spending, V11 = number of health facilities x health social security agency (BPJS) per city or district, H = Hazard, E = Exposure, V = Vulnerability.

## Data Availability

All data generated or analyzed during this study are included in this published article.

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
