# Peer review of "Optimal Validated Multi-Factorial Climate Change Risk Assessment for Adaptation Planning and Evaluation of Infectious Disease: A Case Study of Dengue Hemorrhagic Fever in Indonesia"

_tropicalmed, 2022, doi:10.3390/tropicalmed7080172_

Round 1

Author Response

Thank you.

Reviewer 2 Report

The paper is well written and interesting for the aspects of dengue surveillance in regard to Climate change parameters.

Below are few suggestions for improving the study:

Introduction:

- Please clarify the aim and objective of the study (lines 86-93)

- There are recent studies and systematic reviews on the impacts of urbanisation on dengues, pls consider citing them (eg lines 54 & 318)

Results:

-Line 278 change "syste" to "systems"

-Lines 278 to 280, please consider expanding, comparing, commenting to other global frameworks for supporting Early Warning System interventions such as https://www.who.int/teams/environment-climate-change-and-health/climate-change-and-health/capacity-building/toolkit-on-climate-change-and-health/early-warning-systems

Discussion:

-Please consider expanding the discussion section with analysis on how results can influence adaptation policies such as H-NAPs, also expand on the public health system response to the available data produced by the analysis

Conclusions:

-Lines 414-421 rather belong to the discussion sections as they analyse the results of the study. Please consider either moving them to the discussion section and expand on the policy perspectives produced by the current study, or consider re-phrasing by emphasizing on the conclusions that the current study can provide to scientists, policy makers, public health practitioners and wider audience.

Author Response

Thank you.

Reviewer 3 Report

The work conducted by Faridah et al. is focused on the Optimal validated multi-factorial climate change risk assessment for adaptation planning and evaluation of infectious disease: A Case Study of Dengue Hemorrhagic Fever in Indonesia.

The work is clearly written, the analyses well conducted and the results reported are novel and interesting.

I haven't specific suggestions for the authors.

Author Response

Thank you.

Round 2

Reviewer 1 Report

The authors have adequately addressed my suggestions in the revised version.